# Dynamic right ventricular and atrial volume responses to exercise in endurance-trained and untrained healthy individuals

**Björn Östenson**, **Ellen Ostenfeld**, **Jonathan Edlund**, **Håkan Arheden**, **Katarina Steding-Ehrenborg***

Clinical Physiology, Department of Clinical Sciences Lund, Lund University, Skåne University Hospital, Lund, Sweden

* katarina.steding_ehrenborg@med.lu.se

## Abstract

### Aims

Left ventricular (LV) function is enhanced during exercise in endurance-trained (ET) compared to untrained (UT) healthy individuals. However, the volume responses of the right ventricle (RV), left atrium (LA), and right atrium (RA) during exercise have not been studied in the same subject simultaneously and while standardizing for the respiratory influence on cardiac volumes. The aim was therefore to investigate how the four-chambered heart responds to exercise in ET and UT healthy individuals using exercise real-time cardiac magnetic resonance imaging (CMR) while controlling for effects of respiration.

### Materials and methods

Twenty ET (9 women) and 13 UT healthy individuals matched for age and sex underwent CMR at rest and during moderate and vigorous exercise. LV and RV end-diastolic and end-systolic volumes (EDV and ESV), stroke volumes (SV) and maximal and minimal LA and RA volumes ($LAV_{max}$, $RAV_{max}$ and $LAV_{min}$, $RAV_{min}$) were measured at end expiration.

### Results

LVEDV and LVESV decreased during exercise in both groups. RVEDV and RVESV decreased during exercise in ET but was unchanged in UT. $LAV_{max}$ and $LAV_{min}$ were unchanged from rest to vigorous exercise in both groups. $RAV_{max}$ decreased during exercise in ET whilst $RAV_{min}$ was unchanged, and $RAV_{max}$ and $RAV_{min}$ were unchanged in UT. Thus, the RV and RA volume responses during exercise differed between ET and UT.

**Citation:** Östenson B, Ostenfeld E, Edlund J, Arheden H, Steding-Ehrenborg K (2026) Dynamic right ventricular and atrial volume responses to exercise in endurance-trained and untrained healthy individuals. PLoS One 21(5): e0347745. https://doi.org/10.1371/journal.pone.0347745

**Data availability statement:** All relevant data are within the paper and its Supporting Information files.

**Funding:** This study was funded by the Swedish Research Council for Sport Science (Centrum för idrottsforskning; grant number 2020-0049), the Swedish Olympic Committee, Region Skåne (grant number 2018-Project0123), and Lund University Medical Faculty. All funding was awarded to KSE. The sponsors or funders had no role in study design, data collection and analysis, decision to publish, or preparation of the manuscript.

**Competing interests:** The authors have declared that no competing interests exist.

## Conclusion

This study shows differences between ET and UT healthy individuals in adaptations to exercise when examining the four chambers simultaneously and even when standardizing for respiratory influences. This contributes to increased understanding of the RV volume adaptations in ET.

## Introduction

Endurance-trained (ET) healthy individuals have enlarged left and right ventricular and atrial volumes at rest compared to untrained (UT) healthy individuals [1–3]. This exercise-induced cardiac volume adaptation of the whole four-chambered heart is associated with improved endurance performance [4–6]. However, to our knowledge no previous study has investigated the volume change of all four cardiac chambers in the same subject during exercise, nor assessed if it differs between ET and UT healthy individuals. Studying the whole four-chambered heart during exercise is necessary to fully understand the mechanisms leading to superior cardiac performance in ET compared to UT healthy individuals.

Furthermore, there is a need for increased understanding of the exercise-induced volume adaptations of the right ventricle (RV). As shown by La Gerche et al. [7], the RV is exposed to higher wall stress during exercise compared to the left ventricle (LV). This may affect both long-term adaptations to exercise as suggested by previous studies [8,9] but also how the heart pumps during acute exercise.

Cardiovascular magnetic resonance imaging during exercise (Exercise-CMR) is a challenging method as both the movement of the heart and the respiration may affect image quality. Furthermore, the respiration also affects physiology as it changes intrathoracic pressures and thereby affects cardiac filling [10]. Previous studies have not standardized for the inspiratory and expiratory influences on cardiac volumes during exercise [11,12]. By using a new and validated post-processing algorithm, real-time CMR images can be sorted based on the respiratory phase [13].

Thus, the aim of this study was to investigate the cardiac four-chamber volumetric change from rest to exercise in ET and UT healthy individuals while controlling for effects of respiration.

## Materials and methods

The study was designed as a single center, prospective observational study and was approved by the regional ethics committee in Lund (Dnr 741/2004, with complementary Dnr 269/2005) and performed in concordance with the Declaration of Helsinki. All participants signed informed consent. The study complies with the STROBE guidelines for reporting cross-sectional observational studies [14].

### Study population

Study participants were recruited in 2017–2019 through contact with a local triathlon association, digital and physical advertisements, and word of mouth. ET were

competing in triathlons at club level with training spread across the three training disciplines of swimming, cycling, and running. ET were defined as having peak oxygen consumption ($VO_2$peak) in the upper 95th percentile for their sex and age [15]. UT were consecutively enrolled and matched on group level for age and sex to ET. UT was defined as doing ≤150 minutes of moderate intensity aerobic exercise per week, which is the minimum recommended physical activity by the World Health Organization [16]. Exclusion criteria for all participants were cardiovascular disease, other diseases requiring regular medication such as pulmonary and systemic diseases, smoking, and CMR contraindications.

**Cardiac magnetic resonance imaging and exercise protocol**

CMR was performed using a 1.5 T scanner (Siemens Aera, Siemens Healthineers, Forchheim, Germany). Standard ECG-gated balanced steady-state free precession (bSSFP) images were acquired at rest during breath-hold in a short-axis stack covering the LV, RV, LA, and RA, and in 2-chamber, 3-chamber, and 4-chamber long-axis views with one slice. Typical image parameters were as follows: spatial resolution 1.0 x 1.0 mm, slice thickness 8 mm with no slice gap, temporal resolution 43 ms, flip angle 68˚, echo time 1.1 ms, repetition time 41 ms.

During moderate and vigorous exercise, real-time bSSFP sequences were used to acquire a short-axis stack covering the LV, RV, LA, and RA, and 2-chamber, 3-chamber, and 4-chamber long-axis images with three slices. Typical image parameters for real-time bSSFP were as follows: spatial resolution 1.9 x 1.9 mm, slice thickness 10 mm with no slice gap, temporal resolution 28 ms, flip angle 60˚, echo time 0.9 ms, repetition time 28 ms.

Study participants were scanned in supine position head-first and with feet fastened to the pedals of an MR-compatible cycle ergometer (Lode BV, Groningen, The Netherlands) (Fig 1). Participants were scanned at rest and during moderate and vigorous exercise in the same session – first at rest, then at moderate intensity exercise, and after a short rest (<5 minutes), participants performed an exercise bout at vigorous exercise intensity. Moderate and vigorous exercise intensities were based on individual heart rate (HR) during image acquisition, corresponding to 60 and 80 percent of age-predicted maximum HR, respectively, and defined according to established exercise intensity terminology [17].

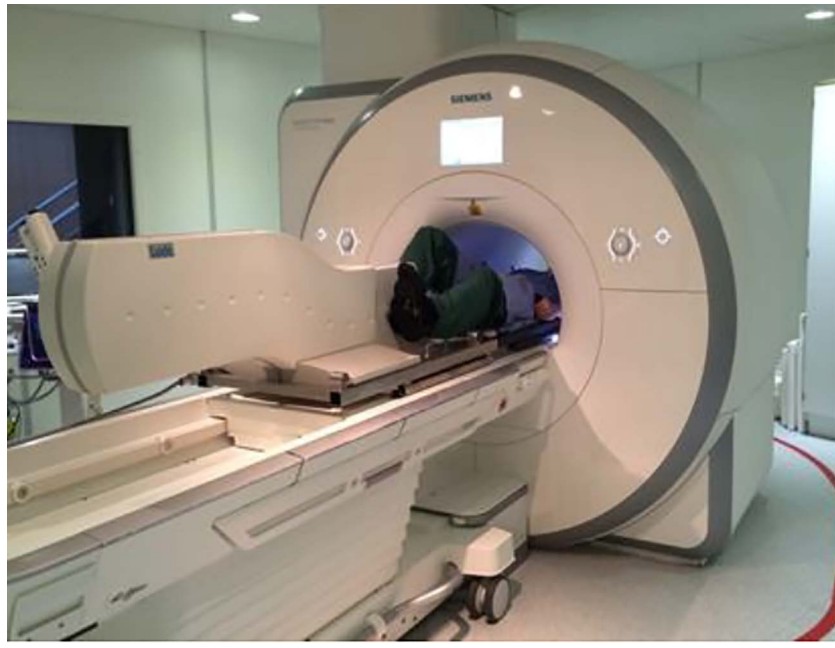

**Fig 1. Experimental setup for cycle ergometer (Lode BV, Groningen, The Netherlands) used for exercise CMR.**

Age-predicted maximum HR was calculated from the formula 220 minus age in years. Starting resistance at both exercise scans were set to 50 W and was manually ramped up to target HR. Images were acquired during fixed exercise resistance.

## Data analysis

Image analysis was performed blinded using the freely available software Segment v3.2 R8688 (Medviso AB, Lund, Sweden, http://segment.heiberg.se) [18]. LV endo- and epicardium and RV endocardium were delineated by one observer (BÖ, eight years of CMR experience) in short-axis images for ventricular end-diastolic (EDV) and end-systolic volumes (ESV) and for LV mass (LVM) (Fig 2). Stroke volume (SV) was calculated as EDV subtracted by ESV. Real-time images were analyzed for the same cardiac variables at end-expiration where the respiratory phase was identified by using a validated semi-automatic sorting algorithm [13] (Fig 3). Measuring in the same respiratory phase reduces beat-to-beat variability in chamber volumes related to intrathoracic pressure changes and improves reproducibility.

Left and right atrial (LA and RA) endocardium were delineated in the short-axis stack by two observers (JE and BÖ, respectively) with similar CMR experiences for volumetric assessment at ventricular end diastole ($LAV_{min}$, $RAV_{min}$) and ventricular end systole ($LAV_{max}$, $RAV_{max}$) (Fig 2). Atrial appendages were included, while pulmonary veins, coronary sinus, and the superior and inferior vena cava were excluded from atrial volumes. LA and RA atrial emptying fractions (LAEF and RAEF) were calculated as the difference between atrial volumes at ventricular end-systole and end-diastole, divided by atrial volume at ventricular end-systole. Absolute volumes for all four chambers (LV, RV, LA, and RA) were indexed by body surface area calculated from the Mosteller formula.

Intra-observer and inter-observer variability was measured in ten participants (six ET and four UT). Of these, two were at rest, four during moderate exercise, and four during vigorous exercise. For intra-observer variability, variables

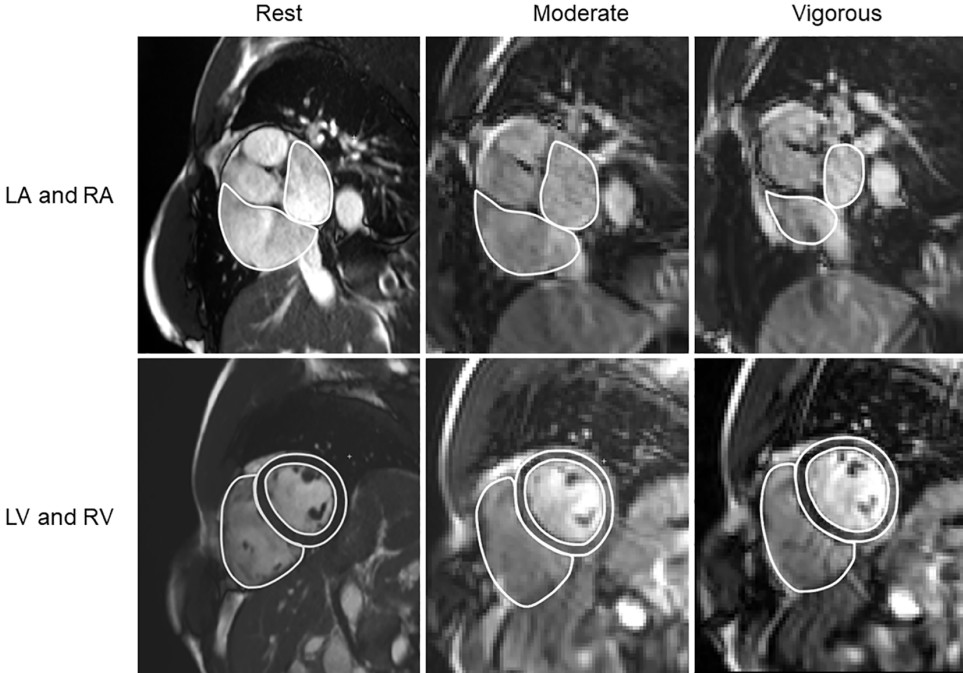

**Fig 2. Delineations of the four cardiac chambers at rest and during exercise in end diastole.** Cardiac magnetic resonance images of the left and right ventricles (LV and RV, top row) and left and right atria (LA and RA, bottom row) at rest (left), moderate exercise (middle), and vigorous exercise (right).

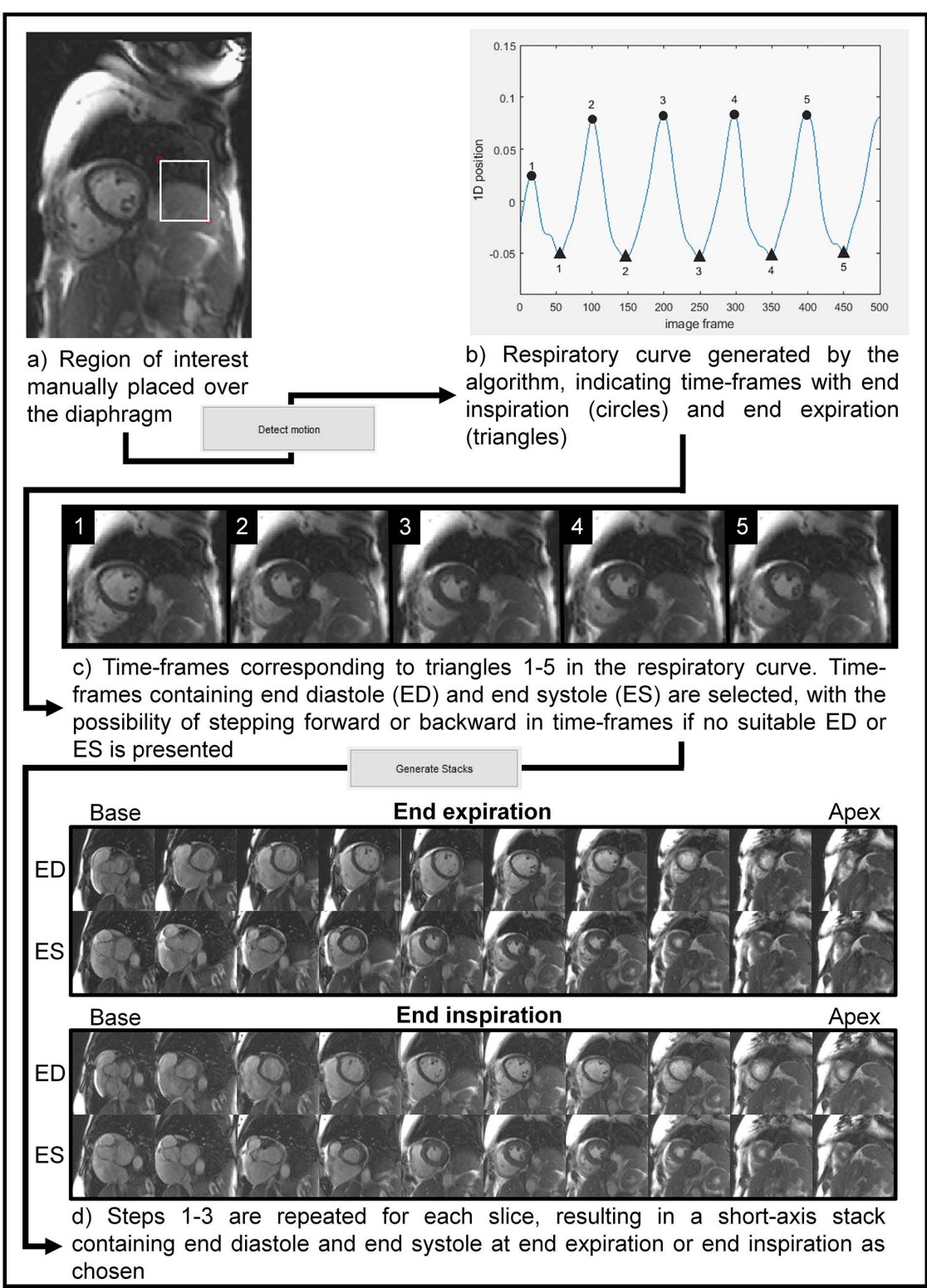

a) Region of interest manually placed over the diaphragm

Detect motion

b) Respiratory curve generated by the algorithm, indicating time-frames with end inspiration (circles) and end expiration (triangles)

c) Time-frames corresponding to triangles 1-5 in the respiratory curve. Time-frames containing end diastole (ED) and end systole (ES) are selected, with the possibility of stepping forward or backward in time-frames if no suitable ED or ES is presented

Generate Stacks

d) Steps 1-3 are repeated for each slice, resulting in a short-axis stack containing end diastole and end systole at end expiration or end inspiration as chosen

**Fig 3. Respiratory sorting algorithm.** End-diastolic and end-systolic short-axis image stacks at end expiration were generated from real-time CMR images according to the flowchart. Adapted from Edlund J, Haris K, Ostenfeld E, Carlsson M, Heiberg E, Johansson S, et al. Validation and quantification of left ventricular function during exercise and free breathing from real-time cardiac magnetic resonance images. Sci Rep. 2022;12: 5611. https://doi.org/10.1038/s41598-022-09366-8, licensed under CC BY 4.0; https://creativecommons.org/licenses/by/4.0/.

were assessed blinded twice with >6 months in between, and for inter-observer variability, blinded by a second observer (KSE, > 15 years of experience).

Resting HR was computed from RR-intervals of cine bSSFP images. During exercise, HR was computed as the mean of systole-to-systole-intervals during 30 heartbeats in real-time bSSFP images, and if less than 30 heartbeats from as many heartbeats as available. Cardiac output (CO) was calculated as LVSV multiplied by HR.

### Cardiopulmonary exercise testing

All participants performed a cardiopulmonary exercise test (CPET) on a stationary bike in upright position at least one day before (n = 17; 9 ± 4 days) or after (n = 16; 8 ± 5 days) the exercise CMR examination. Participants were instructed to not participate in strenuous exercise at least two days before the test and to not consume any products containing caffeine, or a large meal ≥2 hours before the test. Individualized exercise protocols were based on age, and sex, self-reported fitness level according to clinical routine [19]. Incremental resistance was added until exhaustion, or until an even cycling cadence could not be maintained. Maximal cardiopulmonary stress was determined by an overall assessment by the responsible physician, including a respiratory exchange ratio (RER) >1.10, age-predicted maximum HR ≥ 85%, or reached respiratory compensation point indicating cardiopulmonary maximal stress. $VO_2$peak was measured using the equipment Oxycon Champion (Jaeger, Hochberg, Germany) and defined as the mean of the three highest 10-second average $VO_2$ values.

### Statistical analysis

Statistical analyses were conducted using the software IBM SPSS version 28 (SPS inc., Chicago, Illinois, USA) and Graphpad Prism version 10.0.0 for Windows (GraphPad Software, Boston, Massachusetts USA). Minimum sample size needed to detect a medium effect size (Cohen's d = 0.5) with a power of 0.8 and significance level of 0.05 was n = 10, as calculated by the statistical software G*Power 3.1.9.7 (Heinrich-Heine-Universität Düsseldorf, Germany) [20]. Continuous variables are expressed as mean ± SD according to Gaussian distribution. Normal distributions of dependent variables were assessed by Shapiro-Wilk normality test. Discrete variables are expressed as absolute numbers and proportions (in percentages). Independent t-tests were used to compare variables between groups. A two-way mixed model ANOVA with Tukey post-hoc test for multiple comparisons of rest, moderate and vigorous exercise intensities (within-groups factor) and between ET and UT (between-groups factor) was used. The relative change of the dependent variables from rest to moderate and vigorous exercise in ET and UT were compared using unpaired two-samples Student's t-test. Intra-observer and inter-observer variability was expressed as intraclass correlation coefficients (ICC) with 95% confident intervals (CI) and as bias ± 1.96 SD (95% limits of agreement, LoA) according to Bland-Altman [21]. ICC was defined as: < 0.50 poor, 0.50–0.74 adequate, 0.75–0.89 good, and ≥0.90 excellent [22]. Two-tailed P-value of <0.05 was defined as statistically significant.

## Results

Twenty ET (40 ± 10 years, 9 females) and 13 UT healthy individuals matched on group level for age and sex (40 ± 12 years, 7 females) were included. Study participants' baseline characteristics are presented in Table 1. ET had higher $VO_2$peak, and self-reported training volume compared to UT (p < 0.05 for both variables).

### Cardiac volumes and function

All four cardiac chambers were larger in ET compared to UT at rest (Table 2). In ET, LVEDV, RVEDV, LVESV, and RVESV decreased, and LV and RV ejection fraction (LVEF and RVEF) increased from rest to vigorous exercise. For UT, LVEDV and LVESV decreased from rest to vigorous exercise, however RVEDV, RVESV, LVEF and RVEF remained unchanged. (Table 2, Fig 4).

**Table 1. Participants' characteristics.**

|  | UT (n = 13) | ET (n = 20) | p-value |
|---|---|---|---|
| Male / Female (n) | 7 / 6 | 11 / 9 | 0.95 |
| Age (years) | 40 ± 12 | 40 ± 10 | 0.98 |
| Height (cm) | 176 ± 5 | 172 ± 8 | 0.10 |
| Weight (kg) | 69 ± 7 | 67 ± 9 | 0.52 |
| BMI (kg/m$^2$) | 22 ± 2 | 22 ± 2 | 0.57 |
| BSA (m$^2$) | 1.8 ± 0.1 | 1.8 ± 0.2 | 0.34 |
| SBP (mmHg) | 118 ± 16 | 123 ± 13 | 0.44 |
| DBP (mmHg) | 73 ± 10 | 75 ± 8 | 0.68 |
| Training volume (h/week) | 0 ± 1 | 7 ± 3 | <0.001 |
| VO$_2$peak (ml/kg/min) | 38 ± 7 | 54 ± 6 | <0.001 |

Data expressed as mean values ± standard deviation. BMI: body mass index; BPM: beats per minute; BSA: body surface area (calculated using the Mosteller formula); DBP: diastolic blood pressure; ET: endurance-trained; HR: heart rate; SBP: systolic blood pressure; UT: untrained; VO$_2$peak: peak oxygen consumption.

Resting HR was lower in ET than UT, but both groups achieved similar HR when exercising at moderate and vigorous intensity. Thus, ET had a larger increase in HR from rest to exercise (**Table 2**). LVSV was larger in ET than UT at all intensities and was unchanged with exercise in both groups. RVSV was likewise higher in ET, but decreased during vigorous exercise in ET while remaining unchanged in UT. The change in CO with higher exercise intensity was more pronounced in ET compared to UT.

LA volumes and LAEF were unchanged with exercise in both groups (**Table 2**). In contrast, RAV$_{max}$ decreased in ET, while RA volumes were unchanged in UT. RAEF decreased during exercise in both groups.

### Intra-observer and inter-observer variability

The intra-observer variability of the method used with post-processing respiratory sorting of real-time images was excellent with small biases (<5%). The inter-observer variability was excellent for LV and atrial volumes, good for RV volumes, and adequate for LVM. Biases between observers were small for LV and LA volumes (<5%) and substantial for RV volumes and LVM (**Table 3**).

## Discussion

This study assessed the volumes of the four cardiac chambers simultaneously during exercise while standardizing for respiratory effects and showed a difference in cardiac response to exercise between ET and UT for the right side of the heart. ET showed a decrease in both RV volumes and RA$_{max}$, whereas right heart volumes of UT remained unchanged. As the novel method with post-processing respiratory sorting of real-time images was shown reliable in assessing LV, RV, LA, and RA volumes simultaneously upon repeated measures, it can thus beneficially be utilized in further investigations on cardiac exercise physiology.

### Effects of exercise on cardiac volumes and function

The findings from the current study on LV volumes and function are in line with previous exercise CMR studies on healthy populations [23]. While the volume response of the LV with exercise in healthy participants is well studied, the data is scarce for the RV, LA, and RA [24]. However, the current findings of decreased RVEDV and RVESV in ET and unchanged RVEDV and RVESV in UT are in line with previous studies with comparable exercise intensities [25,26].

**Table 2. Left and right ventricular and atrial volumes at rest, during moderate and vigorous exercise intensity in endurance-trained and untrained healthy individuals.**

| | UT | | | ET | | |
|---|---|---|---|---|---|---|
| | **Rest** | **Moderate** | **Vigorous** | **Rest** | **Moderate** | **Vigorous** |
| HR (beats/min) | 60±6 | 128±12# (116±32%) | 151±10#§ (155±34%) | 50±7* | 123±14# (150±39%)* | 152±17#§ (207±43%)* |
| CO (L/min) | 6.0±0.9 | 13.7±2.4# (131±40%) | 15.3±4.3# (155±68%) | 6.3±1.3 | 16.8±2.9*# (173±62%)* | 19.9±2.7*#§ (223±59%)* |
| CI (L/min/m$^2$) | 3.3±0.5 | 7.5±1.3 | 8.3±2.1 | 3.5±0.6 | 9.4±1.6 | 11.2±1.6 |
| **Left ventricle** | | | | | | |
| LVEDV (ml) | 170±32 | 176±34 (4±11%) | 155±32§ (−8±15%) | 215±39* | 215±37* (1±10%) | 199±30*#§ (−6±11%) |
| LVEDVi (ml/m$^2$) | 93±14 | 96±18 | 85±15 | 120±16 | 120±13 | 111±12 |
| LVESV (ml) | 69±19 | 67±19 (−2±14%) | 55±14#§ (−17±25%) | 88±21* | 79±21# (−10±14%) | 67±19#§ (−23±16%) |
| LVESVi (ml/m$^2$) | 38±9 | 37±10 | 30±8 | 49±9 | 44±10 | 38±9 |
| LVSV (ml) | 101±14 | 108±21 (8±16%) | 100±24 (−1±17%) | 127±22* | 137±24* (9±14%) | 132±16* (5±16%) |
| LVSVi (ml/m$^2$) | 55±6 | 59±11 | 54±11 | 71±10 | 76±10 | 74±8 |
| LVEF (%) | 60±4 | 62±6 (4±8%) | 64±7 (8±13%) | 59±4 | 64±6# (8±8%) | 67±6# (13±11%) |
| **Right ventricle** | | | | | | |
| RVEDV (ml) | 180±36 | 182±41 (2±15%) | 164±41 (−8±21%) | 234±42* | 211±34*# (−8±14%) | 193±34*# (−16±21%) |
| RVEDVi (ml/m$^2$) | 98±16 | 99±20 | 89±20 | 131±19 | 118±13 | 108±17 |
| RVESV (ml) | 83±23 | 79±26 (−5±19%) | 70±22 (−12±31%) | 111±23* | 87±19# (−20±20%)* | 82±25# (−22±35%) |
| RVESVi (ml/m$^2$) | 45±11 | 43±13 | 38±11 | 62±11 | 48±8 | 46±13 |
| RVSV (ml) | 97±15 | 103±21 (7±21%) | 93±22 (−3±22%) | 124±21* | 124±22* (2±18%) | 111±15*#§ (−9±14%) |
| RVSVi (ml/m$^2$) | 53±6 | 56±10 | 51±11 | 69±10 | 70±10 | 62±8 |
| RVEF (%) | 54±4 | 57±6 (6±12%) | 58±5 (6±13%) | 53±4 | 59±5# (12±12%) | 58±6# (10±16%) |
| **Left atrium** | | | | | | |
| LAV$_{max}$ (ml) | 76±15 | 92±20# (22±23%) | 85±19 (14±19%) | 101±21* | 110±26*# (9±17%) | 102±24*§ (1±17%) |
| LAVi$_{max}$ (ml/m$^2$) | 41±7 | 50±9 | 46±9 | 57±10 | 61±13 | 57±12 |
| LAV$_{min}$ (ml) | 30±7 | 31±9 (10±46%) | 29±11 (0±45%) | 43±14* | 43±17* (6±49%) | 38±13 § (−7±37%) |
| LAVi$_{min}$ (ml/m$^2$) | 16±3 | 17±4 | 16±6 | 24±7 | 24±9 | 21±8 |
| LAEF (%) | 61±5 | 66±6 (10±15%) | 67±10 (11±19%) | 58±9 | 62±10 (9±23%) | 64±9 (12±23%) |
| **Right atrium** | | | | | | |
| RAV$_{max}$ (ml) | 113±26 | 111±21 (1±17%) | 100±31 (−11±23%) | 144±35* | 129±35# (−10±16%) | 114±35#§ (−20±14%) |
| RAVi$_{max}$ (ml/m$^2$) | 61±12 | 60±9 | 54±15 | 80±17 | 72±17 | 67±15 |
| RAV$_{min}$ (ml) | 56±17 | 64±15 (20±30%) | 63±21 (20±49%) | 76±23* | 76±25 (2±23%) | 68±26 (−10±20%) |
| RAVi$_{min}$ (ml/m$^2$) | 30±9 | 35±8 | 34±11 | 42±12 | 42±12 | 38±13 |

*(Continued)*

**Table 2.** (Continued)

|  | UT | | | ET | | |
|---|---|---|---|---|---|---|
| RAEF (%) | 51 ± 11 | 43 ± 11# (−14 ± 25%) | 37 ± 10# (−22 ± 30%) | 47 ± 7 | 41 ± 8# (−11 ± 22%) | 41 ± 10 (−13 ± 19%) |

Data expressed as mean values ± standard deviation, with relative change compared to rest values in parenthesis. CO: cardiac output; ET: endurance-trained; HR: heart rate; LAEF: left atrial emptying fraction; $LAV_{max}$: maximal left atrial volume; $LAVi_{max}$: maximal left atrial volume index; $LAV_{min}$: minimal left atrial volume; $LAVi_{min}$: minimal left atrial volume index; LVEDV: left ventricular end-diastolic volume; LVEDVi: left ventricular end-diastolic volume index; LVEF: left ventricular ejection fraction; LVESV: left ventricular end-systolic volume; LVESVi: left ventricular end-systolic volume index; LVSV: left ventricular stroke volume; LVSVi: left ventricular stroke volume index; RAEF: right atrial emptying fraction; $RAV_{max}$: maximal right atrial volume; $RAVi_{max}$: maximal right atrial volume index; $RAV_{min}$: minimal right atrial volume; $RAVi_{min}$: minimal right atrial volume index; RVEDV: right ventricular end-diastolic volume; RVEDVi: right ventricular end-diastolic volume index; RVEF: right ventricular ejection fraction; RVESV: right ventricular end-systolic volume; RVESVi: right ventricular end-systolic volume index; RVSV: right ventricular stroke volume; RVSVi: right ventricular stroke volume index; UT: untrained.* $p < 0.05$ ET compared to UT at corresponding intensity # $p < 0.05$ compared with rest § $p < 0.05$ compared with moderate exercise.

The LV response across exercise intensities was the same in ET and UT, while the RV responded differently between groups. As the RV of the athlete's heart has gained increasing attention in the past years in relation to exercise-induced cardiomyopathies [7,9], the healthy athlete's response to exercise is of interest. As previously described by La Gerche et al., high RV wall stress during exercise may result in a relatively high work output and metabolic demand, driving RV adaptations seen in ET but not in UT [7]. Physiologically, the RV differs from the LV in terms of morphology and load. The RV has a thinner wall thickness compared to the LV, and both pre- and afterload is lower in RV than in LV. During exercise, mean pulmonary artery pressure increases around 1.4 mmHg per 1 liter increase in CO [27], resulting in a more than two-fold increase compared to resting values [28]. The systemic pressure, on the other hand, increases only around 60% during endurance exercise [29]. Thus, the wall stress of the RV is disproportionally increased with increased CO and pulmonary artery pressure during exercise [30]. It could be speculated that ET may subject their RV to these high demands repeatedly, driving structural and functional adaptations not seen in UT. The current results show that in health, the RV can withstand the increased load as shown by the unchanged RV volumes in UT and even decreased volumes in ET. In contrast, a dilating RV during exercise may be a marker of an overloaded ventricular wall. This speculation is however beyond the scope of the current study.

The four-chamber volumetric assessment allows the findings to be considered in the context of the constant volume hypothesis of cardiac function, which proposes that total heart volume remains relatively stable [31]. The observed reductions in RV and RA volumes during exercise in the ET group may reflect more efficient redistribution of blood through the pulmonary circulation rather than four-chamber dilation. However, because total heart volume and pulmonary blood volume were not directly quantified, these interpretations remain speculative and should be explored in future studies.

ET had lower RVSV at vigorous exercise intensity than at rest, while the LVSV was unchanged from rest to vigorous exercise. This finding can be explained by the respiratory influence on LV and RV volumes, where LVSV is at its peak during end expiration and RVSV is at its peak during end inspiration [10]. As chamber volumes were quantified at end expiration, a slight difference between LVSV and RVSV was expected as LV and RV volumes have been shown to interact with respiration [10].

It has been discussed that there is a potential adaptation and physiological benefit in decreased RV volumes during moderate exercise as the radial pumping of the RV may be an important factor in diastolic filling [26]. Furthermore, large blood volumes must pass through the RA and RV during exercise, and it can be speculated that in order for the right heart to increase SV, the RV must act less as a reservoir and more as a conduit allowing blood to efficiently continue forward. Both the hypothesis of increased radial pumping enhancing diastolic filling, and a change from acting as a reservoir to a conduit are supported by the current results showing decreased RVEDV and RVESV and are in line with a previous study of the RV during exercise [26]. Other mechanisms explaining decreased RV volumes during exercise are likely the decreased filling time and enhanced contractility [32]. These proposed mechanisms together may be important factors aiding the RV to feed the LV with blood during exercise.

 

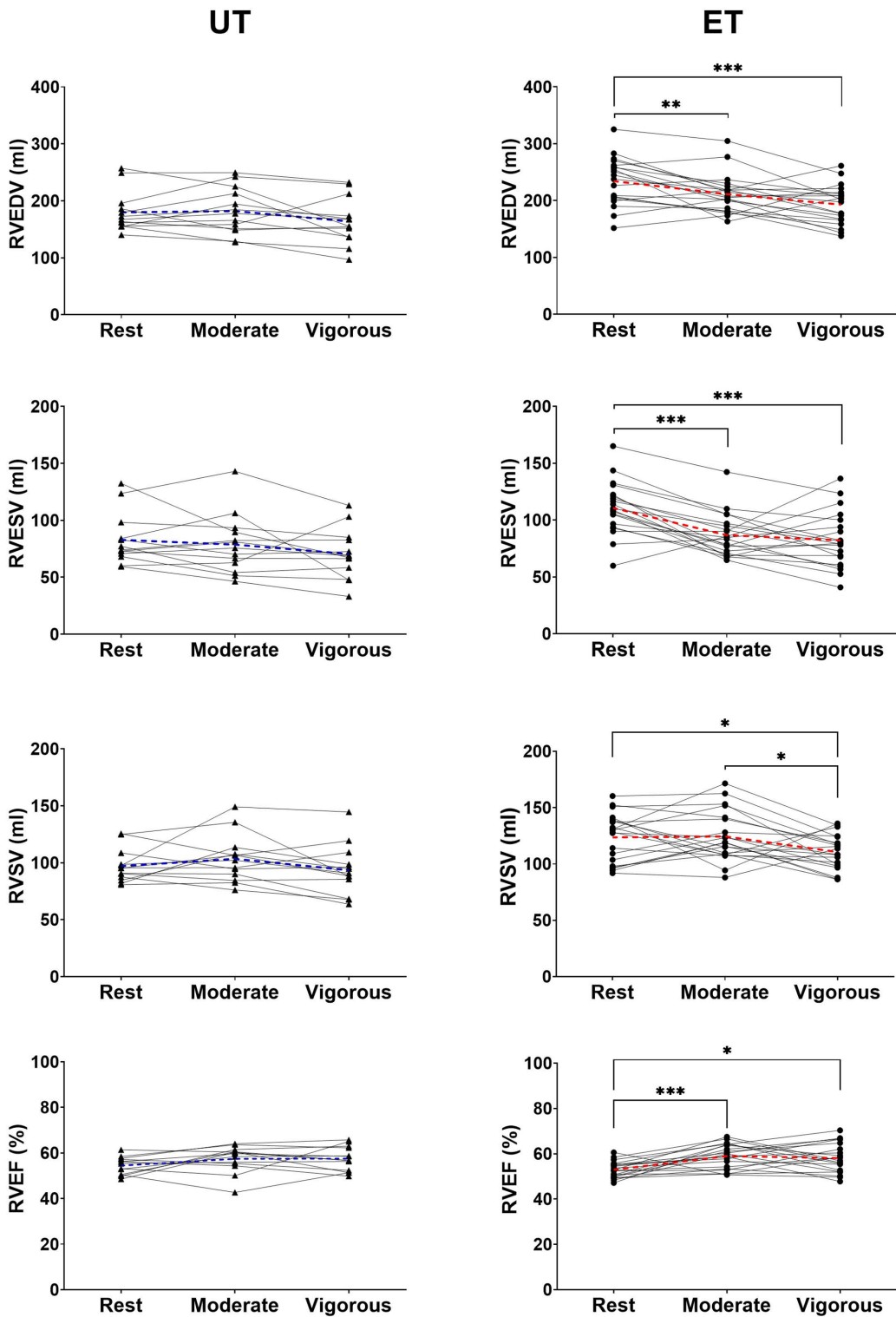

**Fig 4. Right ventricular end-diastolic volume (RVEDV), end-systolic volume (RVESV), stroke volume (RVSV), and ejection fraction (RVEF) in UT (left column) and endurance-trained (ET) healthy individuals (right column) at rest, during moderate and vigorous intensity exercise.** The dashed lines denote mean values in UT (blue) and ET (red). UT had unchanged RVEDV, RVESV, and RVEF whereas ET had decreased RVEDV and RVESV and increased RVEF during exercise. * $p < 0.05$, ** $p < 0.01$, *** $p < 0.001$.

**Table 3. Intra-observer and inter-observer variability.**

| | Intra-observer variability | | | Inter-observer variability | | |
|---|---|---|---|---|---|---|
| | ICC (95% CI) | Bias±95% LoA | | ICC (95% CI) | Bias±95% LoA | |
| | | Absolute | Relative | | Absolute | Relative |
| LVM (g) | 0.91 (0.69–0.98) | 4±18 | 3±14% | 0.60 (−0.06–0.89) | 18±35 | 17±27% |
| LVEDV (ml) | 0.99 (0.95–1.00) | −1±10 | 0±5% | 0.98 (0.93–1.00) | 2±11 | 1±5% |
| LVESV (ml) | 0.96 (0.86–0.99) | −1±10 | −1±12% | 0.90 (0.68–0.98) | −2±17 | −2±22% |
| RVEDV (ml) | 0.99 (0.96–1.00) | 2±10 | 1±5% | 0.87 (0.49–0.97) | 11±31 | 17±27% |
| RVESV (ml) | 0.99 (0.96–1.00) | 1±6 | 1±8% | 0.80 (0.17–0.95) | 12±25 | 21±42% |
| $LAV_{max}$ (ml) | 1.00 (0.99–1.00) | 1±4 | 1±5% | 0.97 (0.88–0.99) | 1±16 | 2±18% |
| $LAV_{min}$ (ml) | 0.98 (0.94–1.00) | 1±4 | 4±18% | 0.93 (0.73–0.98) | 0±10 | −1±42% |
| $RAV_{max}$ (ml) | 1.00 (0.99–1.00) | −1±7 | −1±6% | 0.98 (0.92–0.99) | 1±20 | 10±19% |
| $RAV_{min}$ (ml) | 1.00 (0.99–1.00) | 0±8 | 0±10% | 0.95 (0.84–0.99) | −3±19 | −4±28% |

CI: Confidence interval; ICC: intraclass correlation coefficient; $LAV_{max}$: maximum left atrial volume; $LAV_{min}$: minimum left atrial; LoA: limits of agreement; LVEDV: left ventricular end-diastolic volume; LVESV: left ventricular end-systolic volume; LVM: left ventricular mass; $RAV_{max}$: maximum right atrial volume; $RAV_{min}$: minimum right atrial volume; RVEDV: right ventricular end-diastolic volume; RVESV: right ventricular end-systolic volume.

ET maintained LA and RA function, expressed as total emptying fraction, while UT had slightly lower RA function. The reason for unchanged RA volumes, with a decreased RAEF during exercise in UT is most likely due to statistically non-significant changes in RA volumes. These results are somewhat in conflict with a previous study on LA and RA volumes where LA and RA function increased in ET, non-ET, and patients with chronic thromboembolic pulmonary hypertension during exercise at an intensity of 66% of maximal power output on CPET [12]. One possible explanation for these differences in LA and RA function between studies could be a result of differences in exercise intensity in the studies, where the participants in the present study exercised on a lower intensity (<66% of maximal power output on CPET).

ET and UT had unchanged $LAV_{max}$ and $LAV_{min}$ during exercise, in contrast to the findings in a stress echocardiography study of healthy volunteers where the $LAV_{max}$ and $LAV_{min}$ decreased while LAEF increased [33]. However, the changes in LA volumes were small in both studies, and the differences could thus be a result of statistical power.

## Exercise-induced increase in cardiac output

The results from the present study suggest that CO was increased during exercise mainly by means of increased HR, with only modest changes in SV, especially in UT. There are conflicting results in previous studies regarding the mechanism causing improved cardiac pumping in ET compared to UT during exercise. While some studies have shown that there is a similar pattern of SV increase during exercise with a plateau at higher intensities for both ET and UT [34,35], a recent exercise-CMR review indicate that there is a heterogeneity in SV response to exercise across protocols and cohorts [24]. Differences in the mechanism of increased CO could at least partially be explained by different methods used in previous studies, for example invasive measurements and echocardiography, as well as body position. In the present study, the participants exercised in the supine position and therefore the heart is at a preloaded condition from start. Thus, the "preload reserve" is elevated at rest compared to an upright position, and SV reaches a plateau already at low workloads. At higher workloads, the increased HR reduces diastolic filling time [36], hence limiting SV increase. Furthermore, chamber volumes were assessed at a fixed respiratory phase (end expiration), which means that the full effect of respiration was not captured, as respiration interacts with ventricular volumes [10].

## Limitations

The relatively small sample size of this study could affect statistical power and bias, which may limit generalizability. If underpowered, findings should therefore be considered exploratory and specific to the studied cohort. However, a

priori sample size calculation indicated n = 10 would be sufficient to detect a medium effect size for the primary outcome measures.

The exercise protocol in the present study was based on heart rate, which is a limitation. Although ET and UT were exercised until the same age-predicted maximum HR, the relative effort to reach the same target HR may differ between ET and UT, as ET start from a lower resting HR. Thus, ET need to use a relatively greater proportion of their heart rate reserve to reach target HR. Furthermore, exercising at 60% and 80% of age-predicted maximum HR could result in different absolute and relative workloads in ET and UT. Unfortunately, other means to assess exercise intensity like exercise duration, workload, and oxygen consumption ($VO_2$) at each intensity during CMR were not collected in the present study and can therefore not be compared between groups. Nevertheless, relative HR and relative workload are comparable as measures of exercise intensity as they are linearly associated [37].Image parameters differed between rest and exercise. The temporal resolution of CMR images during exercise is low, making image acquisition at high heart rates challenging. However, before designing the imaging protocol for the current study, real-time images were validated against ECG-gated cine bSSFP images have been validated against real-time images with a low bias [13].

Cardiac volume measurements were restricted to end-expiration, resulting in respiratory-dependent fluctuations in stroke volume were not captured. Thus, the reported volumes reflect standardized rather than fully dynamic cardiorespiratory interactions during exercise.

Total heart volume and pulmonary blood volume variations were not measured. Consequently, potential redistribution of blood between the RV and RA to the LV and LA or from the pulmonary circulation during exercise cannot be directly evaluated.

Although the intra-observer variability of measurements was excellent, the inter-observer variability was less so. The difference between intra-observer and inter-observer variability reflects the challenges of delineating exercise CMR images when temporal and spatial resolutions are limited due to motion artifacts. Test-retest variability was not explored in the current study. However, the intra-observer and inter-observer agreements were excellent when developing the respiratory-sorting algorithm [13], indicating that the method is reliable. Nevertheless, the respiratory-sorting algorithm was developed for the LV, and not the RV with its more complex shape. This, and that the RV is more challenging to delineate, may partly explain the higher variabilities in the RV volumes.

A limitation of this study is that cardiac remodeling was assessed using volumetric measures only. Myocardial mass and wall thickness were not evaluated, which limits conclusions regarding structural remodeling beyond dynamic chamber volume responses during exercise.

## Perspective

ET undergo regular training to stimulate physiological adaptations, ultimately leading to enhanced sports-specific performance. The well-known cardiac morphological and functional adaptations in ET observed in previous studies were confirmed in the present study. However, in contrast to previous studies, the present study used the gold standard method of CMR with a novel respiratory-sorting algorithm to evaluate all four chambers simultaneously during exercise in the same respiratory phase. Our findings that the RV and RA volumes decrease during exercise in ET, but not in UT, imply cardiac adaptations to regular training that have not previously been evaluated. Thus, the present study contributes to an increased understanding of the physiological adaptations explaining enhanced sports-specific performance in ET. Future studies may focus on the adaptations of the RV and RA in ET to improve the understanding of the enhanced performance and predict outcomes.

## Conclusion

This study shows differences in volumetric response between ET and UT during exercise when examining the four cardiac chambers simultaneously and when standardizing for respiratory influences. RV volumes decreased, and RV ejection

fraction increased in ET during exercise, whilst it remained unchanged in UT. Similarly, RAV$_{max}$ decreased in ET and remained unchanged in UT. However, for the LV both groups responded similarly with decreased volumes from rest to vigorous exercise, and for the LA volumes remained unchanged during exercise in both groups. This contributes to increased understanding of the cardiac response to exercise in ET and UT, especially for the right side of the heart. The novel method with post-processing respiratory sorting of real-time images was shown reliable in assessing the four-chambered heart upon repeated measures, it can thus beneficially be utilized in further investigations on cardiac exercise physiology.

## Supporting information

**S1 Table. Individual subject data.** CO: cardiac output; ET: endurance-trained; HR: heart rate; LAEF: left atrial emptying fraction; LAVmax: maximal left atrial volume; LAVmin: minimal left atrial volume; LVEDV: left ventricular end-diastolic volume; LVEF: left ventricular ejection fraction; LVESV: left ventricular end-systolic volume; LVSV: left ventricular stroke volume; RAEF: right atrial emptying fraction; RAVmax: maximal right atrial volume; RAVmin: minimal right atrial volume; RVEDV: right ventricular end-diastolic volume; RVEF: right ventricular ejection fraction; RVESV: right ventricular end-systolic volume; RVSV: right ventricular stroke volume; SBP: systolic blood pressure; UT: untrained.
(XLSX)

**S2 Table. Individual subject data for intraobserver and interobserver analysis.** ET: endurance-trained; LAVmax: maximal left atrial volume; LAVmin: minimal left atrial volume; LVEDV: left ventricular end-diastolic volume; LVESV: left ventricular end-systolic volume; LVM: left ventricular mass; RAVmax: maximal right atrial volume; RAVmin: minimal right atrial volume; RVEDV: right ventricular end-diastolic volume; RVESV: right ventricular end-systolic volume; RVESVi: right ventricular end-systolic volume index; UT: untrained.
(XLSX)

## Acknowledgments

The authors would like to thank Sebastian Bidhult for contributing to data acquisition.

Current address of authors: Department of Clinical Physiology, Skane University Hospital in Lund, Entrégatan 7, 22185 Lund, Sweden

## Author contributions

**Conceptualization:** Katarina Steding-Ehrenborg.

**Data curation:** Björn Östenson, Katarina Steding-Ehrenborg.

**Formal analysis:** Björn Östenson.

**Funding acquisition:** Katarina Steding-Ehrenborg.

**Investigation:** Björn Östenson.

**Resources:** Katarina Steding-Ehrenborg.

**Supervision:** Ellen Ostenfeld, Katarina Steding-Ehrenborg.

**Validation:** Björn Östenson, Jonathan Edlund, Katarina Steding-Ehrenborg.

**Visualization:** Björn Östenson.

**Writing – original draft:** Björn Östenson.

**Writing – review & editing:** Björn Östenson, Ellen Ostenfeld, Jonathan Edlund, Håkan Arheden, Katarina Steding-Ehrenborg.

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
