## [Decision Letter · Decision Letter 0]

8 Jan 2026

PONE-D-25-57809Right ventricular and atrial volumes decrease in athletes but not in healthy controls during exercisePLOS One

Dear Dr. Östenson,

Thank you for submitting your manuscript to PLOS ONE. Thank you for submitting your manuscript to PLOS ONE. Two external referees have evaluated it, and the consensus is that it is of interest to the journal; however, there are several issues that merit a major revision. Therefore, we invite you to submit a revised version of the manuscript that addresses the points raised during the review process.

If applicable, we recommend that you deposit your laboratory protocols in protocols.io to enhance the reproducibility of your results. Protocols.io assigns your protocol its own identifier (DOI) so that it can be cited independently in the future. For instructions see: https://journals.plos.org/plosone/s/submission-guidelines#loc-laboratory-protocols. Additionally, PLOS ONE offers an option for publishing peer-reviewed Lab Protocol articles, which describe protocols hosted on protocols.io. Read more information on sharing protocols at . Additionally, PLOS ONE offers an option for publishing peer-reviewed Lab Protocol articles, which describe protocols hosted on protocols.io. Read more information on sharing protocols at https://plos.org/protocols?utm_medium=editorial-email&utm_source=authorletters&utm_campaign=protocols..

We look forward to receiving your revised manuscript.

Kind regards,

Neftali Eduardo Antonio-Villa, MD PhD

Academic Editor

PLOS One

Journal Requirements:

3. In the online submission form, you indicated that the datasets generated and analysed during the current study are not publicly available due to the small sample size and the risk of identifying individual participants but are available from the corresponding author on reasonable request. Data are available from the Cardiac MR Group in Lund, Lund University, Sweden for researchers who meet the criteria for access to confidential data and after additional consent from the research participants. For data access requests, please contact the Cardiac MR Group in Lund via email: cmr-lund@med.lu.se..

5. We note that Figure(s) 1, 2, 3, in your submission contain copyrighted images. All PLOS content is published under the Creative Commons Attribution License (CC BY 4.0), which means that the manuscript, images, and Supporting Information files will be freely available online, and any third party is permitted to access, download, copy, distribute, and use these materials in any way, even commercially, with proper attribution. For more information, see our copyright guidelines: http://journals.plos.org/plosone/s/licenses-and-copyright.

a. You may seek permission from the original copyright holder of Figure(s) 1, 2, 3, to publish the content specifically under the CC BY 4.0 license.

Additional Editor Comments :

- Please carefully addressed the comments raised by two external referees.

- Emphasize in the potential clinical implications and consider in adding the limitations raised by the reviewers.

Reviewers' comments:

Reviewer's Responses to Questions

**Comments to the Author**

1. Is the manuscript technically sound, and do the data support the conclusions?

Reviewer #1: Partly

Reviewer #2: Partly

2. Has the statistical analysis been performed appropriately and rigorously? 

Reviewer #1: Yes

Reviewer #2: Yes

3. Have the authors made all data underlying the findings in their manuscript fully available?

Reviewer #1: No

Reviewer #2: Yes

4. Is the manuscript presented in an intelligible fashion and written in standard English?

Reviewer #1: Yes

Reviewer #2: Yes

5. Review Comments to the Author

Reviewer #1: Thank you for the opportunity to review this paper.

This is a descriptive MRI based study of cardiac chamber volumes during exercise, comparing athletes with non-exercising controls.

1. The population appears well controlled for age and sex distribution which is a strength. The authors might consider a description of groups as endurance trained versus untrained healthy individuals; the information provided does not describe competition level, years of training, or a distinction for whether the athletes are recreationally competitive versus elite.

2. Performance of CMR with exercise obviously requires selected expertise, which is also a strength of this study.

3. Strictly speaking, the concept of cardiac chamber remodeling should encompass more than a measure of volumes at end diastole and end systole, and might include more information about chamber mass, or ratio of chamber morphology to cavity dimensions. I would consider removing the comment in the abstract conclusion, and perhaps elsewhere that the study provides understanding of RV remodeling.

4. The paper does contribute carefully measured, chamber volumes from a healthy population, for which more information is clearly welcome. The ability to control for respiratory phase was also an interesting strength. I would have found it additionally helpful for my understanding of this information, given the age range and sex distribution, if the volume data could also be presented indexed for body morphology.

5. It was curious that with information from a CPET study, that moderate and vigorous stages were defined by a percentage of the crude formula for max HR. It appears that some subjects had peak VO2 measured after the imaging study. It would also contextualize the findings to also show the %peak VO2 achieved at each stage, to understand if exercise stages were comparable between the 2 groups.

6. As per point 3, rather than true chamber remodeling, the paper provides more information about dynamic volume alterations that produce ejection based on intensity of exercise in trained and untrained individuals. Considering this, the presentation of data at a single phase of the respiratory cycle seemed incomplete to understand how the volume data contribute to the performance of the heart under exercising conditions.

a. As the authors discuss on page 17, the data as presented suggest that, in both groups, the primary contributor to increases in CO was the change in HR. Although I would agree, the literature conflict somewhat, to see no change in LV SV or LVEF in the control group would not be expected at all based on hemodynamic studies of exercise. The restricted respiratory phase evaluation may then restrict us from seeing the whole picture, and perhaps presenting the inspiratory phase data, or the data from changes in volumes and LV or RVEF may help us contextualize the data better.

b. As the paper presents data from all 4 chambers, do the authors subscribe to the constant volume hypothesis of heart function? Is there a shift in RV cardiac output and LV cardiac output contributions during exercise? Such information may help us understand if higher blood volume sequestration in the pulmonary circulation contributes to differences between groups.

Reviewer #2: Reviewer Recommendation: Minor Revision

1. Summary of the study

This study evaluates the physiological adaptations of the four heart chambers during exercise in athletes compared to healthy controls, utilizing real-time cardiac magnetic resonance (CMR) imaging. The researchers employ a novel post-processing respiratory sorting method to assess atrial and ventricular volumes while minimizing respiratory influences.

2. General Comments

The manuscript addresses a relevant and interesting topic in cardiac physiology. The methodology is well-executed, particularly the standardization for respiratory influences and the use of a reliable post-processing sorting method. This provides a solid technical framework. The paper is well-written, although some minor adjustments regarding the scope and the presentation of the sample size are needed to strengthen the final version.

3. Points for Revision

Sample Size and Statistical Considerations: While the study provides valuable insights, the total sample size (N=33) is relatively small. The authors should briefly address the potential limitations regarding statistical power and risk of bias in the Discussion section. It is important to clarify that these findings are exploratory and specific to this cohort, rather than fully extrapolatable to the broader athletic population.

4. Title Refinement: To prevent overgeneralization, I strongly recommend revising the title to more accurately reflect the study’s specific parameters. Given the limited sample size (N=33), the current title implies a universal physiological adaptation that may not be representative of athletic populations globally. I suggest delimiting the title by specifying both the precise cohort of athletes studied and the geographical/regional context where the data was collected (e.g., 'Right ventricular and atrial volumes in a specific cohort of [Type] athletes in [City/Country]: An exploratory study'). Incorporating the geographical location is essential, as environmental, genetic, and regional training factors can influence cardiac physiology. This adjustment will ensure that the findings are interpreted within their correct clinical and regional framework.

5.Refining Text and Redundancy: The manuscript would benefit from a more concise presentation. There are instances of redundancy where certain points are reiterated throughout the sections. I recommend a minor editorial review to eliminate these repetitions and improve the flow of the text.

6. References: Please ensure that all scientific claims are supported by their respective citations. A few areas would benefit from updated references (within the last 5 years) to ensure the study is framed within the most current literature in the field.

6. PLOS authors have the option to publish the peer review history of their article (what does this mean?). If published, this will include your full peer review and any attached files.). If published, this will include your full peer review and any attached files.

.

Reviewer #1: No

Reviewer #2: No

---

## [Author Response · Author response to Decision Letter 1]

20 Feb 2026

Detailed point-by-point responses to the editor’s and reviewers’ comments are provided in the attached document "Response to Reviewers."

---

## [Decision Letter · Decision Letter 1]

8 Apr 2026

Dynamic right ventricular and atrial volume responses to exercise in endurance-trained and untrained healthy individuals

PONE-D-25-57809R1

Dear Dr. Östenson,

We’re pleased to inform you that your manuscript has been judged scientifically suitable for publication and will be formally accepted for publication once it meets all outstanding technical requirements.

An invoice will be generated when your article is formally accepted. Please note, if your institution has a publishing partnership with PLOS and your article meets the relevant criteria, all or part of your publication costs will be covered. Please make sure your user information is up-to-date by logging into Editorial Manager at Editorial Manager® and clicking the ‘Update My Information' link at the top of the page. For questions related to billing, please contact  and clicking the ‘Update My Information' link at the top of the page. For questions related to billing, please contact billing support..

Kind regards,

Neftali Eduardo Antonio-Villa, MD PhD

Academic Editor

PLOS One

Additional Editor Comments (optional):

I would like to congratulate the authors on their efforts in addressing all the comments raised by the reviewers. The manuscript is now suitable for publication.

Reviewers' comments:

Reviewer's Responses to Questions

**Comments to the Author**

1. If the authors have adequately addressed your comments raised in a previous round of review and you feel that this manuscript is now acceptable for publication, you may indicate that here to bypass the “Comments to the Author” section, enter your conflict of interest statement in the “Confidential to Editor” section, and submit your "Accept" recommendation.

Reviewer #1: All comments have been addressed

2. Is the manuscript technically sound, and do the data support the conclusions?

Reviewer #1: Yes

3. Has the statistical analysis been performed appropriately and rigorously? 

Reviewer #1: Yes

4. Have the authors made all data underlying the findings in their manuscript fully available?

Reviewer #1: Yes

5. Is the manuscript presented in an intelligible fashion and written in standard English?

Reviewer #1: Yes

6. Review Comments to the Author

Reviewer #1: Thank you for your attention to the reviewer's comments. The manuscript reflects the expertise of the investigators and care of the data.

7. PLOS authors have the option to publish the peer review history of their article (what does this mean?). If published, this will include your full peer review and any attached files.). If published, this will include your full peer review and any attached files.

.

Reviewer #1: No

---

## [Editor Report · Acceptance letter]

PONE-D-25-57809R1

PLOS One

Dear Dr. Östenson,

I'm pleased to inform you that your manuscript has been deemed suitable for publication in PLOS One. Congratulations! Your manuscript is now being handed over to our production team.

Kind regards,

on behalf of

Dr. Neftali Eduardo Antonio-Villa

Academic Editor

PLOS One